# Catastrophic health expenditure: A comparative analysis of smoking and non-smoking households in China

Zhigang Zhong[1]◉, Han Wei[1]◉, Lian Yang[2]*, Tingting Yao[3], Zhengzhong Mao[4], Qun Sun[1]

1 School of Management, Chengdu University of Traditional Chinese Medicine, Chengdu, Sichuan Province, China, 2 School of Public Health, Chengdu University of Traditional Chinese Medicine, Chengdu, Sichuan Province, China, 3 School of Nursing, Institute for Health and Aging, University of California, San Francisco, California, United States of America, 4 Huaxi School of Public Health, Sichuan University, Chengdu, Sichuan Province, China

◉ These authors contributed equally to this work.
* yyanglian@163.com

**Data Availability Statement:** Data for this study are derived from the China Health and Retirement Longitudinal Study (CHARLS). CHARLS is a national, large-scale follow-up project launched in 2011 and tracked once every two years that

## Abstract

### Introduction

Smoking is hazardous to health and places a heavy economic burden on individuals and their families. Clearly, smoking in China is prevalent since China is the largest consumer of tobacco in the world. Chinese smoking and nonsmoking households were compared in terms of the incidence and intensity of Catastrophic Health Expenditures (CHEs). The factors associated with catastrophic health expenditures were analyzed.

### Methods

Data for this study were collected from two waves of panel data in 2011 and 2013 from the national China Health and Retirement Longitudinal Study (CHARLS). A total of 8073 households with at least one member aged above 45 were identified each year. Catastrophic health expenditure was measured by the ratio of a household's out-of-pocket healthcare payments (OOP) to the household's Capacity to Pay (CTP). A panel logit random-effects model was used to examine correlates with catastrophic health expenditure.

### Results

The incidence of catastrophic health expenditures for Chinese households with members aged 45 and above in 2011 and 2013 were 12.99% and 15.56%, respectively. The mean gaps (MGs) were 3.16% and 4.88%, respectively, and the mean positive gaps (MPGs) were 24.36% and 31.40%, respectively. The incidences of catastrophic health expenditures were 17.41% and 20.03% in former smoking households, 12.10% and 15.09% in current smoking households, and 12.72% and 13.64% in nonsmoking households. In the panel logit regression model analysis, former smoking households (OR = 1.444, P<0.001) were more prone to catastrophic health expenditures than nonsmoking households. Risk factors for catastrophic health expenditures included members with chronic diseases (OR = 4.359,

presents a high quality nationally representative sample of Chinese residents ages 45 and older. The authors had no special access privileges in accessing data from CHARLS. Data are available by application at (http://charls.pku.edu.cn/index/en. html). It should be noted that the CHARLS database requires a registration application before it can be downloaded and used. Data application form link: (http://charls.pku.edu.cn/pages/data/ 111/en.html) Data user registration link: (http:// charls.pku.edu.cn/users/sign_up/agreement/en. html) Questions can be directed to the CHARLS investigation team (contact via email at charls_info @pku.edu.cn or by phone: 86-400-610-1866 or 86-(0)10-62767425).

**Funding:** This work was supported by National Natural Science Foundation of China, grant no. 71603032.The funders had no role in study design, data collection and analysis, decision to publish, or preparation of the manuscript.

**Competing interests:** The authors have declared that no competing interests exist.

**Abbreviations:** CHE, Catastrophic Health Expenditure; CHARLS, China Health and Retirement Longitudinal Study; OOP, out-of-pocket healthcare payments; CTP, Capacity to Pay; GDP, Gross Domestic Product; PPS, Probabilities Proportional to Size; MG, Mean gap; MPG, Mean positive gap.

P<0.001), hospitalized patients (OR = 8.60, P<0.001), elderly people aged above 65 (OR = 1.577, P<0.001), or persons with disabilities (OR = 1.275, P<0.001). Protective factors for catastrophic health expenditures included being in an urban household, having a larger family size, and having a higher household income.

## Conclusions

The incidence of catastrophic health expenditures in Chinese households is relatively high. Smoking is one of the primary risk factors for catastrophic health expenditures. Stronger interventions against smoking should be made in time to reduce the occurrence of health issues caused by smoking and the financial losses for individuals, families and society.

## Introduction

Hazards resulting from tobacco consumption are one of the most detrimental public health issues in the world today. China, a middle-income country, is the world's largest manufacturer and consumer of tobacco. China's tobacco production makes up 40% of the world's total, and its tobacco consumption accounts for one-third of the world's total. Moreover, the number of Chinese smokers makes up one-third of the world's smoking population[1–3]. According to the "2015 China Adult Tobacco Survey" conducted by the Chinese Center for Disease Control and Prevention, the smoking prevalence of adults is 27.7% in China, and the number of smokers in China is 316 million[4]. The harmful health effects of smoking have been confirmed by numerous studies. Smokers are at increased risk of developing cancer, cardiovascular disease and chronic respiratory illness[5–9]. From 1990 to 2010, the number of deaths caused by smoking increased from 700 thousand to 1.4 million in merely twenty years[10]. Smoking-attributable deaths per year in China are predicted to reach 3 million by 2050 if the problem remains unchecked[11]. Smoking has become the second most serious health risk factor affecting the number of deaths and the reduction of life expectancy in China[12].

According to previous studies, smoking has immense negative effects on the social economy. In 2012, the world's total medical expenditure for smoking-attributable diseases reached 467 billion US dollars, accounting for 5.7% of the global health expenditure. Meanwhile, the total economic cost of smoking (including medical expenditures and productivity losses) in 2012 was 1852 billion US dollars, accounting for 1.8% of the global gross domestic product (GDP). Almost 40% of the economic cost occurs in low-income or middle-income countries[13]. In China, the cost of smoking-attributable diseases in 2008 was 28.9 billion US dollars, accounting for 3.0% of the total health expenditure or 0.7% of GDP in the same year [14].

However, the effects of smoking on household financial status have been sparsely studied. In human society, families are the basic units of the social structure. In the current situation regarding China, an individual is most likely to overcome financial risks caused by disease related issues with the help of his/her family. Smoking is hazardous to health and may result in enormous medical expenses beyond the limit that a standard household can afford. Health expenditures are considered to be catastrophic when a household has to reduce its basic expenditures in order to afford medical expenses[15,16].

Catastrophic Health Expenditure(CHE) can reflect not only whether families have fallen into a catastrophic situation due to excessive health-care costs but also the equity of health financing[17]. There have been many studies on CHE, but most Chinese scholars have focused

on CHE related to chronic noncommunicable diseases, such as hypertension and diabetes[18–21]. So, does smoking incur a heavy economic burden of disease to households? What is the relationship between smoking and CHE?

While attempting to analyze the differences between Chinese smoking and nonsmoking households in the incidence and intensity of CHE, this study aims to explore the influencing factors of CHE. The results will help to reveal the household economic risks brought about by smoking and provide basic information for the Chinese government to formulate tobacco control policies.

## Materials and methods

### Data source

Our research data are mainly derived from the China Health and Retirement Longitudinal Study (CHARLS)[22]. CHARLS is a national, large-scale follow-up project launched in 2011 and tracked once every two years. The survey used a multistage and Probabilities Proportional to Size(PPS) sampling strategy to collect data from 28 provinces/municipalities/autonomous regions in the country (provinces/municipalities/autonomous regions including Hainan Province, Ningxia Autonomous Region, Taiwan Province, and Tibet Autonomous Region were not sampled). The survey subjects are households with members aged 45 and above. With a large sample size covering most parts of the country and different aspects, the CHARLS survey questionnaire is able to provide all the data needed to calculate CHE. Having retained individual and household data tracked in 2011 and 2013, we successfully retrieved balance panel data covering 8073 households with a follow-up rate of 78.74%.

### Methods

In previous studies, there have been two measurement criteria for CHE. When out-of-pocket healthcare payments (OOP) accounted for more than 10% of total household expenditures, CHE was considered to have occurred[23–25]. In addition, when OOP accounted for 40% or more of the household's Capacity to Pay (CTP), the health expenditure was also considered catastrophic[16,18,26–29]. This study adopts the second criterion. A household's CTP is defined as the effective income after basic subsistence needs are satisfied. In many studies, effective income is considered to be the total household consumption expenditure, and basic subsistence needs refer to household food expenditures[16,30]. Therefore, the household's CTP is equal to the result of total household consumption subtracted by household food expenditures[31–33].

The frequency and severity of CHE are often measured with the incidence and intensity of CHE. The incidence of CHE refers to the percentage of households with CHE in all households. The mean gap (MG) and mean positive gap (MPG) reflect the intensity of CHE[18–20]. The MG evaluates the severity of CHE in all sample households, while the MPG measures the severity of CHE in households with CHE. $E_i$ is used to represent whether a household has CHE. The formula is as follows:

$$E_i = 1 \text{ if } \frac{oop_i}{ctp_i} \geq 0.4 \tag{1}$$

$$E_i = 0 \text{ if } \frac{oop_i}{ctp_i} < 0.4 \tag{2}$$

$oop_i$ represents the OOP of the "i-th" family; $ctp_i$ represents the capacity of the "i-th" family to pay. The incidence and intensity of CHE are calculated as follows[34,35]:

$$H_{cat} = \frac{1}{N} \sum_{i=1}^{N} E_i \tag{3}$$

$$MG_{cat} = \frac{1}{N} \sum_{i=1}^{N} E_i \left( \frac{oop_i}{ctp_i} - 0.4 \right) \tag{4}$$

$$MPG_{cat} = \frac{MG_{cat}}{H_{cat}} \tag{5}$$

N represents the number of sample households; $H_{cat}$ represents the incidence of CHE; $MG_{cat}$ represents the MG in CHE, and $MPG_{cat}$ represents the MPG in CHE.

Smoking variables: the households are categorized into nonsmoking households and smoking households, which includes former smoking households and current smoking households. The category of households is determined by the smoking status of the respondents, grouped as nonsmokers, current smokers and former smokers. A respondent is classified as a smoker if he/she answered "yes" to the question "Have you ever chewed tobacco, smoked a pipe, smoked self-rolled cigarettes, or smoked cigarettes/cigars". Current smoker is the one who answered "still have" to the question "Do you still have the habit or have you totally quit" and former smoker is the one whose answer of the same question was "quit". A household with at least one current smoker is a current smoking household, and a household with neither a current smoker nor a former smoker is deemed a nonsmoking household; otherwise, it is a former smoking household.

## Demographic characteristics

Control variables: household size (1–2 people, 3–4 people, ≥5 people), rural vs. urban residence(classified according to the residential area), if a household has chronic disease family member (yes, no), or hospitalized family member (yes, no), or elderly people aged 65 and above (yes, no), or disabled family members (yes, no), or alcoholic members (yes, no), or medical insurance covered members(yes, no), household income categories(poorest, poorer, middle, richer and richest), economic region (western regions: Sichuan, Chongqing, Guizhou, Yunnan, Gansu, Guangxi, Inner Mongolia Autonomous Region, Qinghai, Shaanxi and Xinjiang Uygur Autonomous Region; the central regions: Shanxi, Jilin, Heilongjiang, Anhui, Henan, Hubei, Hunan and Jiangxi; the eastern regions: Beijing, Tianjin, Hebei, Liaoning, Fujian, Guangdong, Jiangsu, Shandong, Shanghai and Zhejiang), and survey year (2011, 2013).

## Statistical analysis

This study adopted the panel Logit regression model to analyze the influencing factors of CHE. The dependent variable is whether a household suffered from CHE. The model is as follows:

$$Logit(Y_{it}) = \beta_0 + \beta_1 Smoking_{it} + \beta_2 X_{it} + \alpha_i + \mu_{it} \tag{6}$$

In this formula, $Y_{it}$ indicates whether household i had CHE in year t. If a household suffered from CHE, then Y = 1; otherwise, Y = 0. $Smoking_{it}$ is the dummy variable of the smoking status; $X_{it}$ is a vector of social demographic characteristics of households. $\alpha_i$ indicates the

unobservable special effect among individuals, and $\mu_{it}$ is a white noise error.i stands for the sample household, while t stands for the year.

A two-tailed p value of <0.05 was considered statistically significant. All data in this study were analyzed with STATA (version 14.0, MP).

## Results

Table 1 shows summary statistics for independent variables in 2011 and 2013. There were 5042 urban households (62.46%) and 3031 rural households (37.54%). From 2011 to 2013, the number of nonsmokers and current smoking households decreased, whereas the number of former smoking households increased. The percentages of households with family members with chronic diseases, hospitalized family members, members covered by medical insurance or elderly people aged 65 and above also increased. The numbers of households with 3–4 members and households with 5 or more members increased. The percentage of households with disabled members decreased.

From 2011 to 2013, all figures, including household consumption expenditures, food expenditures, the CTP, and OOP, showed upward trends. By 2013, the average household consumption expenditures had reached $ 4,242.94, whereas food expenditures had reached $1,892.38. The household's ability to pay had reached $2632.57, and out-of-pocket health payments had reached $373.93. Among different categories of smoking households, the consumption expenditures of former smoking households were the highest in 2013 at $5,214.94, and the nonsmoking households' expenditures were the lowest at $4,191.71. The household out-of-pocket health payments were the highest among former smoking households at $572.07. The OOP of current smoking households and nonsmoking households were relatively close at $336.33 and $320.97, respectively. Details are shown in Table 2.

Table 3 shows that the incidences of CHE for all households in 2011 and 2013 were 12.99% and 15.56%, respectively. From 2011 to 2013, the MG increased from 3.16% to 4.88%, and the MPG increased from 24.36% to 31.40%. In 2011, the incidence of CHE in former smoking households was the highest (17.41%). There was only a slight difference between the incidences in current smoking and nonsmoking households, which were 12.10% and 12.72%, respectively. In 2013, the incidence of CHE in former smoking households was up to 20.03%, and the incidence of CHE in current smoking households was 15.09%, which was 1.45% higher than that in nonsmoking households. In both 2011 and 2013, the highest MG existed among former smoking households (15.09%), the second-highest MG was found among the current smoking households, and the lowest MG was found among the nonsmoking households.

Fig 1 presents the incidences of CHE in different households and income levels in 2011 and 2013 at different CHE thresholds (20%, 30%, 40%, 50%, and 60%). Consistent trends over the two years were evident. The smaller the threshold was, the higher the incidence of household CHE. At the same threshold, former smoking households were more prone to CHE, and the incidences in nonsmoking households and current smoking households were similar, but the gap widened over time (Fig 1A and 1B). The trend was consistent over the two years, and as income increased, the incidence of CHE decreased (Fig 1C and 1D).

As shown in Table 4, former smoking households and households with chronically ill members, hospitalized members, or elderly people aged 65 and above were more likely to have CHE. Urban households were more likely to avoid CHE than rural households. The larger the household size, the less likely it was to have CHE. More affluent families were less prone to CHE.

## Discussion

The results from this study showed that the incidences of CHE in Chinese households in 2011 and 2013 were 12.99% and 15.56%, respectively. Using data from China's Fourth National

**Table 1. Descriptive statistics of independent variables in 2011 and 2013[N/(%)].**

| Independent variable | 2011 | 2013 |
|---|---|---|
| Household category | | |
| Non smoking household | 2610(32.33) | 2309(28.60) |
| Current smoking household | 4412(54.65) | 4326(53.59) |
| Former smoking household | 1051(13.02) | 1438(17.81) |
| Residence | | |
| Urban area | 5042(62.46) | 5042(62.46) |
| Rural area | 3031(37.54) | 3031(37.54) |
| Having chronic disease members in household | | |
| No | 1010(12.51) | 684(8.47) |
| Yes | 7063(87.49) | 7389(91.53) |
| Having hospitalized members in household | | |
| No | 6908(85.57) | 6375(78.97) |
| Yes | 1165(14.43) | 1698(21.03) |
| Having medical insurance covered members in household | | |
| No | 370(4.58) | 240(2.97) |
| Yes | 7703(95.42) | 7833(97.03) |
| Having elderly people aged 65 and above in household | | |
| No | 5811(71.98) | 5311(65.79) |
| Yes | 2262(28.02) | 2762(34.21) |
| Having disabled members in household | | |
| No | 5992(74.22) | 6558(81.23) |
| Yes | 2081(25.78) | 1515(18.77) |
| Having alcoholic members in household | | |
| No | 4642(57.50) | 4647(57.56) |
| Yes | 3431(42.50) | 3426(42.44) |
| Household size | | |
| 1–2 persons | 3312(41.03) | 2910(36.05) |
| 3–4 persons | 2635(32.64) | 2863(35.46) |
| ≥5 persons | 2126(26.33) | 2300(28.49) |
| Household income level | | |
| Q1 | 1623(20.10) | 1616(20.02) |
| Q2 | 1608(19.92) | 1616(20.02) |
| Q3 | 1614(19.99) | 1618(20.04) |
| Q4 | 1617(20.03) | 1609(19.93) |
| Q5 | 1611(19.96) | 1614(19.99) |
| Location | | |
| The west | 2734(33.87) | 2734(33.87) |
| The middle | 2634(32.63) | 2634(32.63) |
| The east | 2705(33.51) | 2705(33.51) |

Health Service Survey (2008), Li Y et al. found that the incidence of CHE was 13.0%[36]. In another study by the same team, the incidence of CHE in rural Chinese households was 14.4% [37]. Meng et al. found that the incidences of Chinese household CHE in 2003, 2008, and 2011 were 12.2%, 14.0% and 12.9%, respectively[38]. Wenjuan et al. utilized the same data source as used in this study (2015 CHARLS national data) and discovered that the incidence of Chinese household CHE was 16.5%[39]. Our study mainly focused on Chinese households with middle-aged and senior members aged 45 and above, but the results were similar to those of the

**Table 2. Chinese household payment capacity and OOP in 2011 and 2013 (USD).**

| | 2011 | | 2013 | |
|---|---|---|---|---|
| | **Mean** | **Standard deviation** | **Mean** | **Standard deviation** |
| **Non-smoking households** | | | | |
| Household consumption expenditure | 3560.11 | 5037.48 | 4119.71 | 6060.45 |
| Household food expenditure | 1442.47 | 1771.68 | 1605.35 | 2115.38 |
| Household payment capacity | 2117.64 | 4460.06 | 2514.36 | 5201.61 |
| Household out of pocket health payment | 246.98 | 961.77 | 320.97 | 913.84 |
| **Current smoking households** | | | | |
| Household consumption expenditure | 3792.1 | 5436.81 | 4541.79 | 7370.18 |
| Household food expenditure | 1750.01 | 2602.53 | 2053.02 | 3729.73 |
| Household payment capacity | 2042.09 | 4423.79 | 2488.78 | 5838.04 |
| Household out of pocket health payment | 218.87 | 545.43 | 336.33 | 942.06 |
| **former smoking households** | | | | |
| Household consumption expenditure | 3843.88 | 5206.89 | 5124.94 | 9305.41 |
| Household food expenditure | 1463.22 | 1561.07 | 1870.00 | 3437.63 |
| Household payment capacity | 2380.66 | 4762.03 | 3254.95 | 8210.69 |
| Household out of pocket health payment | 399.64 | 1176.57 | 572.07 | 1567.25 |
| **Total** | | | | |
| Household consumption expenditure | 3723.84 | 5281.51 | 4524.94 | 7425.14 |
| Household food expenditure | 1613.25 | 2248.41 | 1892.38 | 3297.61 |
| Household payment capacity | 2110.59 | 4481.68 | 2632.57 | 6171.18 |
| Household out of pocket health payment | 251.49 | 803.16 | 373.93 | 1077.09 |

whole population study. One possible reason for the similar results might be that, according to life cycle theory, an individual's economic consumption is not determined by current disposable income, and rational investors in families plan their consumption and property income in their lifetimes. In other words, they usually work hard to establish sufficient savings accounts when they are young and have an abundance of savings at hand after retirement[40]. Middle-aged and senior people have more health care expenditures, but they also have more savings and are more impervious to family financial risks.

Our study found that the incidence of CHE in former smoking households was much higher than that of current and nonsmoking households in different years and at different thresholds. Similarly, the MG and MPG were also larger in former smoking households. A possible reason is that the health hazards of smoking lag behind the time spent smoking. Health damage usually occurs 10 to 20 years after the first exposure to tobacco[41], and smoking-related diseases may not occur or be detected in current smokers. In addition, some studies report that current smokers are less concerned about their health and are therefore less likely to seek medical care[42]. Finally, most users are unaware of the risks of tobacco use. Many smokers in China will not take the initiative to quit smoking until diseases are

**Table 3. Incidence and intensity of CHE in different Chinese smoking households in 2011 and 2013 (%).**

| | 2011 | | | 2013 | | |
|---|---|---|---|---|---|---|
| | **Incidence** | **MG** | **MPG** | **Incidence** | **MG** | **MPG** |
| Non-smoking households | 12.72 | 2.85 | 22.43 | 13.64 | 4.25 | 31.16 |
| Current smoking households | 12.10 | 3.10 | 25.59 | 15.09 | 4.69 | 31.06 |
| Former smoking households | 17.41 | 4.22 | 24.26 | 20.03 | 6.49 | 32.42 |
| Total | 12.99 | 3.16 | 24.36 | 15.56 | 4.88 | 31.40 |

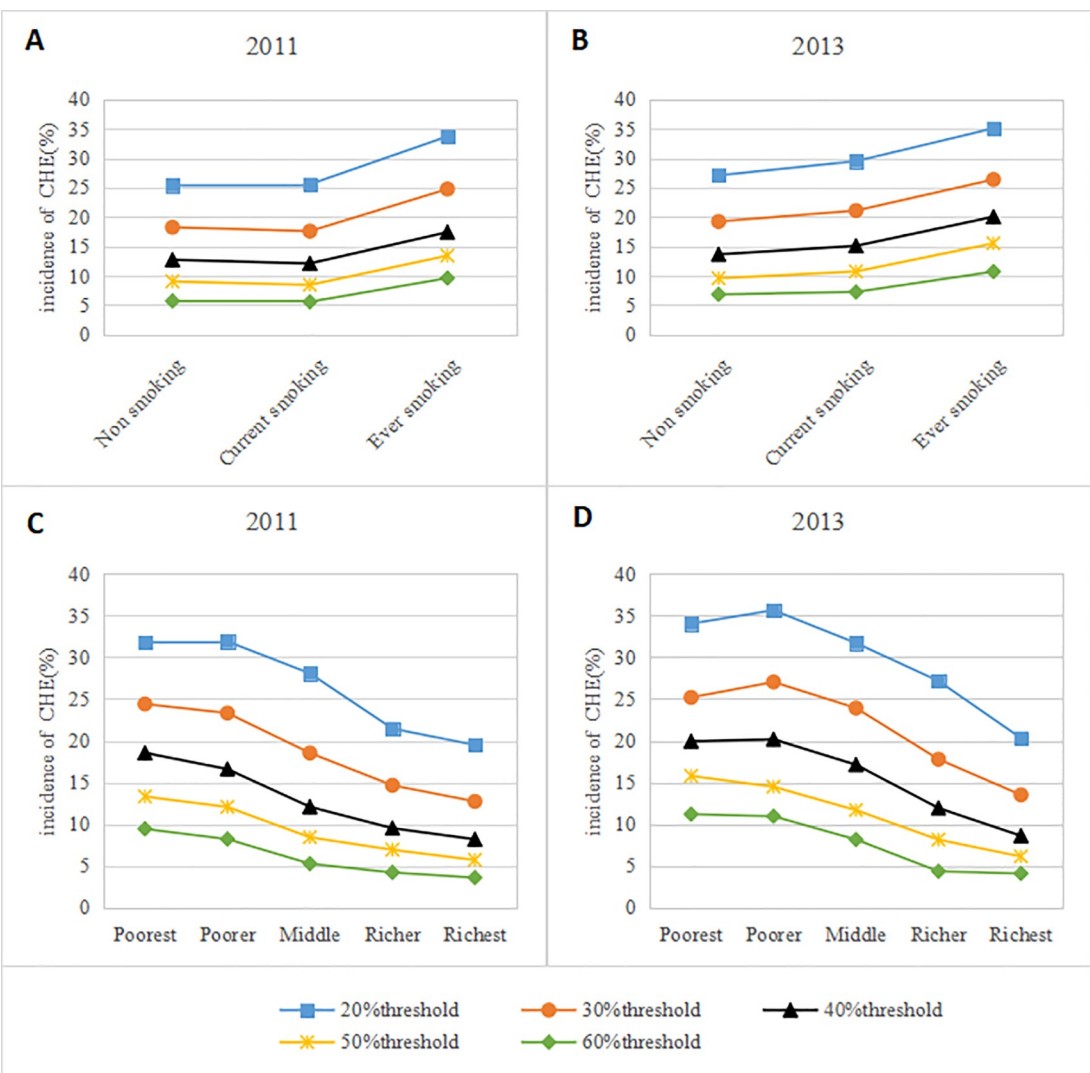

**Fig 1.** Incidence of CHE among different Chinese smoking households (A and B) and income levels (C and D)at different thresholds in 2011 and 2013.

diagnosed[43,44]. Only 17.7% of smokers planned to quit smoking within the next year according to the 2015 China Adult Tobacco Survey[4].

As is consistent with previous research results, lower income households were more likely to have CHE[18,20,29,36]. This finding may be due to the relatively low purchasing power in low-income families. As most medical services are necessity goods, low purchasing power does not prevent a family from paying for medical expenses when it needs medical services. This necessity may result in medical expenses that exceed a family's purchasing power, thus incurring CHE[45]. Additionally, households living in rural areas were more likely to suffer from CHE than households living in urban areas due to their differences in income levels. In 2013, the Chinese per capita disposable income of urban residents was three times that of rural residents[46].

Similar to previous studies[18–21,36,45], our study showed that households with chronically ill patients or disabled or hospitalized members were more likely to have CHE than healthy households. On the one hand, poor health conditions lower labor work capacity and

**Table 4. Panel logit random effects model results of influencing factors of Chinese household CHE.**

| Independent variables | β | SE | Wald | P | OR | 95%CI |
|---|---|---|---|---|---|---|
| Household type | | | | | | |
| Non-smoking household | | | | | 1 | |
| Current smoking household | 0.99 | 0.073 | 1.36 | 0.174 | 1.104 | 0.957–1.273 |
| Former smoking household | 0.367 | 0.091 | 4.06 | <0.001 | 1.444 | 1.209–1.724 |
| Residence | | | | | | |
| Rural area | | | | | 1 | |
| Urban area | -0.221 | 0.069 | -3.19 | 0.001 | 0.801 | 0.670–0.918 |
| Having chronic disease members in household | | | | | | |
| No | | | | | 1 | |
| Yes | 1.472 | 0.158 | 9.33 | <0.0001 | 4.359 | 3.199–5.938 |
| Having hospitalized members in household | | | | | | |
| No | | | | | 1 | |
| Yes | 2.152 | 0.074 | 29.16 | <0.0001 | 8.6 | 7.442–9.938 |
| Having medical insurance covered members in household | | | | | | |
| No | | | | | 1 | |
| Yes | 0.118 | 0.162 | 0.73 | 0.466 | 1.125 | 0.819–1.547 |
| Having elderly people aged 65 and above in household | | | | | | |
| No | | | | | 1 | |
| Yes | 0.455 | 0.066 | 6.94 | <0.0001 | 1.577 | 1.386–1.793 |
| Having disabled members in household | | | | | | |
| No | | | | | 1 | |
| Yes | 0.243 | 0.066 | 3.7 | <0.0001 | 1.275 | 1.121–1.450 |
| Having alcoholic members in household | | | | | | |
| No | | | | | 1 | |
| Yes | -0.125 | 0.063 | -1.98 | 0.048 | 0.882 | 0.779–0.999 |
| Household size | | | | | | |
| 1–2 persons | | | | | 1 | |
| 3–4 persons | -0.875 | 0.074 | -11.88 | <0.0001 | 0.417 | 0.361–0.482 |
| ≥5 persons | -1.333 | 0.084 | -15.91 | <0.0001 | 0.264 | 0.224–0.311 |
| Household income level | | | | | | |
| Q1 | | | | | 1 | |
| Q2 | -0.074 | 0.085 | -0.88 | 0.381 | 0.929 | 0.787–1.096 |
| Q3 | -0.289 | 0.09 | -3.22 | 0.001 | 0.749 | 0.628–0.893 |
| Q4 | -0.469 | 0.094 | -4.99 | <0.0001 | 0.626 | 0.520–0.752 |
| Q5 | -0.652 | 0.101 | -6.48 | <0.0001 | 0.521 | 0.428–0.635 |
| Location | | | | | | |
| The west | | | | | 1 | |
| The middle | -0.006 | 0.075 | -0.08 | 0.938 | 0.994 | 0.859–1.151 |
| The East | -0.093 | 0.077 | -1.21 | 0.225 | 0.911 | 0.783–1.059 |
| **Year** | | | | | | |
| 2011 | | | | | 1 | |
| 2013 | 0.063 | 0.056 | 1.13 | 0.258 | 1.065 | 0.955–1.188 |

production efficiency, which limits a family's economic output and reduces its income. On the other hand, poor health results in higher demand for medical services and incurs medical expenditures. In particular, some serious diseases have even caused heavy economic burden, adding to the risk of household CHE.

In summary, we have the following policy recommendations. First, the Chinese government should curb the prevalence of tobacco consumption in China and encourage smokers to quit smoking early. The Healthy China 2030 plan clearly states that the smoking prevalence in China should be reduced to 20% by 2030[12,47]. We should take more actions to highlight smoking hazards. For example, nationwide educational campaigns could be organized to educate the public about the dangers of tobacco use, and smoke-free ambassadors could be nominated for their "star effect". Moreover, health warnings on all tobacco product packages should be made mandatory. Above all, increasing the tobacco tax is likely to be the most effective way to reduce tobacco use[48]. In 2015, China's second increase in tobacco tax caused a slight increase in tobacco prices, but the reform had little impact on tobacco sales volume[49]. The increase in household income exceeds the increase in cigarette prices. It is recommended that the Chinese government raise the tobacco tax again, allowing for benefits to the country on all fronts[50]. Second, more financial protection to households at risk of CHE should be provided. With the expansion of basic social medical insurance coverage in China, the proportion of out-of-pocket medical expenses to total medical expenses has dropped from 56% in 2003 to 34% in 2013[51], but the medical insurance system has not effectively reduced CHE yet[52,53]. This issue is due to the lack of effective financial protection from the health insurance system to vulnerable groups, including people who are poor, chronically ill or disabled. A series of measures should be taken, such as expanding the coverage of catastrophic medical insurance and medical aid, increasing the scope of insurance reimbursement for chronic disease outpatient services, and improving reimbursement for inpatient services to provide more support to vulnerable groups.

There are also some limitations in this study. First, the household out-of-pocket health payment data used in this study only included direct medical expenses and excluded indirect morbidity expenses (such as transportation and caregivers, absence from work). This conservative estimation method may lead to an underestimation of household CHE. Moreover, as CHARLS is a retrospective self-reported survey, recall bias may be inevitable.

## Conclusions

Tobacco has a significant impact on CHE in Chinese households. Former smoking households are more likely to have CHE. Measures should be taken to increase the publicity of tobacco hazards and to urge smokers to quit smoking as soon as possible to reduce the negative consequences of tobacco for individuals, families and society.

## Acknowledgments

We would like to thank Professor Teh-wei Hu, Professor Anita H. Lee and Professor Shuang Ma for providing expert advice.

## Author Contributions

**Conceptualization:** Lian Yang.

**Data curation:** Zhigang Zhong, Han Wei.

**Formal analysis:** Zhigang Zhong, Han Wei, Lian Yang.

**Methodology:** Zhigang Zhong, Han Wei, Lian Yang.

**Project administration:** Qun Sun.

**Resources:** Lian Yang.

**Supervision:** Tingting Yao, Zhengzhong Mao.

**Writing – original draft:** Zhigang Zhong.

**Writing – review & editing:** Lian Yang.

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
