## [Decision Letter · Decision Letter 0]

24 Feb 2020

PONE-D-20-01101

Catastrophic health expenditure: A comparative analysis of smoking and non-smoking households in China

PLOS ONE

Dear Dr. Yang,

Thank you for submitting your manuscript to PLOS ONE. After careful consideration, we feel that it has merit but does not fully meet PLOS ONE’s publication criteria as it currently stands. Therefore, we invite you to submit a revised version of the manuscript that addresses the points raised during the review process.

The revised version should take into account all comments.

We would appreciate receiving your revised manuscript by Apr 09 2020 11:59PM. To enhance the reproducibility of your results, we recommend that if applicable you deposit your laboratory protocols in protocols.io, where a protocol can be assigned its own identifier (DOI) such that it can be cited independently in the future. For instructions see: http://journals.plos.org/plosone/s/submission-guidelines#loc-laboratory-protocols

We look forward to receiving your revised manuscript.

Kind regards,

Petri Böckerman

Academic Editor

PLOS ONE

Journal Requirements:

"This study was supported by the National Natural Science Foundation of China, grant no. 71603032."

4. Please remove your figures from within your manuscript file, leaving only the individual TIFF/EPS image files, uploaded separately.  These will be automatically included in the reviewers’ PDF.

Reviewers' comments:

Reviewer's Responses to Questions

**Comments to the Author**

1. Is the manuscript technically sound, and do the data support the conclusions?

Reviewer #1: Yes

Reviewer #2: Partly

2. Has the statistical analysis been performed appropriately and rigorously? 

Reviewer #1: Yes

Reviewer #2: No

3. Have the authors made all data underlying the findings in their manuscript fully available?

Reviewer #1: Yes

Reviewer #2: No

4. Is the manuscript presented in an intelligible fashion and written in standard English?

Reviewer #1: Yes

Reviewer #2: Yes

5. Review Comments to the Author

Reviewer #1: Overall, this manuscript provides an estimate of the incidence and severity of catastrophic financial impact on households from tobacco use in China. This is particularly interesting given the large total number of tobacco users in China, which contributes substantially to the global burden of disease. This survey is notable for its high response rate and broad geographic range regarding regions in China. I was surprised to see that current smoker households do not have a substantially higher incidence or proportion of catastrophic health expenditures than nonsmoker households, and some of my recommendations for the manuscript may provide better insight into that finding.

1) it would be helpful to have a copy editor review the manuscript to clarify minor issues with the English wording. It was also not easy to understand the difference between mean gap and mean positive gap - the formulas helped but it would be good to reframe these in the text so that it is clear the mean gap is the proportion of CHE for the group analyzed, and the mean positive gap is the proportion of CHE for those experiencing CHE within the group analyzed.

2) it could be clearer which regions are sampled. Instead of naming four regions not included in the sampling frame on page 12, it would be easier to state the 28 regions (first all listed on page 15)

3) I am not totally clear if this is a pooled regression analysis rather than a panel regression analysis that uses the respondents that were in both waves of the survey. I am not sure what was the random effect? I would have thought it was time but then it appeared in the table.

4) As the unit of analysis is a household, a change in a single individual could be counted the same way as 5+ individuals. It may be helpful to see a sensitivity analysis where the costs are adjusted for individuals in the family rather than just using it as a fixed effect. This may help explain the surprising findings that households with current smokers do not have higher costs compared to nonsmoking households.

5) One of the advantages of panel data is to evaluate effects among individuals undergoing transitions. it would be helpful if the study took advantage of the panel data available. Instead of analyzing only 3 categories (non, current, former) there could be analyses of seven categories based on 2011 and 2013 status (non to non, current to current, former to former, non to current, non to former, current to former, former to current). Given the reported rate of missingness, it is unclear how many people are transitioning from one state (nonsmoking/former/current) to another (nonsmoking/former/current). Also understanding these transitions would help explain why nonsmoking and current smoking households both decline between 2011 and 2013 which seems unusual as nonsmoking households can only become smoking households or stay nonsmoking, and not feed into the former category (unless allowed to have someone smoke for a year and then quit before the 2013 survey. The transition states may also help clarify the surprising findings for current smokers, as those who were current smokers in 2011 may have different trajectories for those in 2013 that become former smokers vs those that stay current smokers. Is it that groups transitioning from current to former have the highest proportion and severity of CHE?

6) I am unclear what is a meaningful difference for Table 3.

Reviewer #2: For my review of ``Catastrophic health expenditure: A comparative analysis of smoking and non-smoking households in China'' Please see attachment, which explains my responses to the questions above.

6. PLOS authors have the option to publish the peer review history of their article (what does this mean?). If published, this will include your full peer review and any attached files.

Reviewer #1: No

Reviewer #2: Yes: Christopher F Baum

---

## [Author Response · Author response to Decision Letter 0]

20 Apr 2020

Funding Statement：This work was supported by National Natural Science Foundation of China, grant no. 71603032.The funders had no role in study design, data collection and analysis, decision to publish, or preparation of the manuscript.

---

## [Decision Letter · Decision Letter 1]

13 May 2020

Catastrophic health expenditure: A comparative analysis of smoking and non-smoking households in China

PONE-D-20-01101R1

Dear Dr. Yang,

We are pleased to inform you that your manuscript has been judged scientifically suitable for publication and will be formally accepted for publication once it complies with all outstanding technical requirements.

With kind regards,

Petri Böckerman

Academic Editor

PLOS ONE

Additional Editor Comments (optional):

Reviewers' comments:

Reviewer's Responses to Questions

**Comments to the Author**

1. If the authors have adequately addressed your comments raised in a previous round of review and you feel that this manuscript is now acceptable for publication, you may indicate that here to bypass the “Comments to the Author” section, enter your conflict of interest statement in the “Confidential to Editor” section, and submit your "Accept" recommendation.

Reviewer #1: All comments have been addressed

Reviewer #2: All comments have been addressed

2. Is the manuscript technically sound, and do the data support the conclusions?

Reviewer #1: Yes

Reviewer #2: (No Response)

3. Has the statistical analysis been performed appropriately and rigorously? 

Reviewer #1: Yes

Reviewer #2: (No Response)

4. Have the authors made all data underlying the findings in their manuscript fully available?

Reviewer #1: Yes

Reviewer #2: (No Response)

5. Is the manuscript presented in an intelligible fashion and written in standard English?

Reviewer #1: Yes

Reviewer #2: (No Response)

6. Review Comments to the Author

Reviewer #1: Thank you for the revisions. Only minor issue is that for formula 2 for MPG I think the N over the summation sign should be N2 not N1. Second sentence of the Discussion should change "Current and Former..." to "Current and former..."

Reviewer #2: (No Response)

7. PLOS authors have the option to publish the peer review history of their article (what does this mean?). If published, this will include your full peer review and any attached files.

Reviewer #1: No

Reviewer #2: Yes: Christopher F Baum

---

## [Editor Report · Acceptance letter]

20 May 2020

PONE-D-20-01101R1 

Catastrophic health expenditure: A comparative analysis of smoking and non-smoking households in China 

Dear Dr. Yang:

I am pleased to inform you that your manuscript has been deemed suitable for publication in PLOS ONE. Congratulations! Your manuscript is now with our production department. 

With kind regards,

on behalf of

Professor Petri Böckerman 

Academic Editor

PLOS ONE